# Chemophenetic Approach to Selected Senecioneae Species, Combining Morphometric and UHPLC-HRMS Analyses

**DOI:** 10.3390/plants12020390

**Published:** 2023-01-14

**Authors:** Yulian Voynikov, Vessela Balabanova, Reneta Gevrenova, Dimitrina Zheleva-Dimitrova

**Affiliations:** 1Department of Chemistry, Faculty of Pharmacy, Medical University, 2 Dunav Str., 1000 Sofia, Bulgaria; 2Department of Pharmacognosy, Faculty of Pharmacy, Medical University, 2 Dunav Str., 1000 Sofia, Bulgaria

**Keywords:** *Senecio*, *Jacoboea*, Orbitrap, chemophenetic, clustering, R programming

## Abstract

Herein, a chemophenetic significance, based on the phenolic metabolite profiling of three *Senecio* (*S. hercynicus, S. ovatus*, and *S. rupestris*) and two *Jacobaea* species (*J. pancicii* and *J. maritima*), coupled to morphometric data, is presented. A set of twelve morphometric characters were recorded from each plant species and used as predictor variables in a linear discriminant analysis (LDA) model. From a total 75 observations (15 from each of the five species), the model correctly assumed their species’ membership, except for 2 observations. Among the studied species, *S. hercynicus* and *S. ovatus* presented the greatest morphological similarity. A phytochemical profiling of phenolic specialized metabolites by UHPLC-Orbitrap-MS revealed 46 hydroxybenzoic, hydroxycinnamic, and acylquinic acids and their derivatives, 1 coumarin and 21 flavonoids. Hierarchical and PCA clustering applied to the phytochemical data corroborated the similarity of *S. hercynicus* and *S. ovatus*, observed in the morphometric analysis. This study contributes to the phylogenetic relationships between the tribe Senecioneae taxa and highlights the chemophenetic similarity/dissimilarity of the studied species belonging to *Senecio* and *Jacobaea* genera.

## 1. Introduction

The tribe Senecioneae (Asteraceae) encompasses more than 150 genera and 3000 species; approximately half of its species belong to the genus *Senecio* L., considering it one of the largest genera of flowering plants [1]. *Senecio* species have a wide distribution and they occur in various habitats—from low altitudes to high mountain communities, and from Arctic regions to hot tropical areas [1]. Although phylogenetic studies have been carried out classifying the taxa, the intergeneric relations are still vague [1,2]; some *Senecio* species have been recently transferred to a separate genus, *Jacobaea* Mill. [3]. Within the genus *Senecio*, hybridization was observed, e.g., *S. hercynicus* × *S. ovatus* [4]. Most taxa in the tribe can be identified by the existence of capitula (flower heads) with a typically uniseriate involucre. However, some species are poorly differentiated morphologically, and there is still uncertainty about recognition of their taxa [2,5,6,7]. *Senecio* species are reported to accumulate sesquiterpenoids (eremophilanes, furanoeremophilanes, cacalols, eudesmanes, oplopanes, germacranes, etc.), pyrrolizidine alkaloids (PAs) [1,8,9], phenolic compounds [10,11,12,13,14,15], and various other secondary metabolites [4,9,16]. *Senecio* species have been described to possess analgesic [17] and hypoglycemic [18] activity, related to the typical for the genus sesquiterpene lactones, and insecticidal properties [19] related to the presence of PAs. Additionally, the taxa are reported to express strong antioxidant, cytotoxic, and antimicrobial activity attributed to the presence of phenolic compounds [11,12,13,15].

In the Bulgarian flora, *Senecio hercynicus* Herborg., *S. ovatus* (G. Gaertn. and Al.) Willd., *S. rupestris* Waldst. and Kit., and *Jacobaea pancicii* (Degen) Vladimirov and Raab-Straube are perennial plants distributed in the mountain regions, up to 1500 (2200) m a.s.l., while *J. maritima* (L.) Pelser and Meijden is a shrub spread on the Black Sea coast [20]. Presently, *S. hercynicus* and *S. ovatus* are included in the *S. nemorensis* group. Formerly, *S. hercynicus* was recognized as *S. nemorensis* L.; *S. ovatus* as *Jacobaea ovata* G. Gaertn.; *J. maritima* as *Senecio maritima* (L.) Rchb.; *J. pancicii* as *Senecio pancicii* Degen [20]. Phytochemical studies on some *Senecio* and *Jacobaea* species with Bulgarian origin were based on the characterization of PAs [19,21,22], phenolic, and flavonoid derivatives [14]. Although the studied species are distributed in other European floras, including the floras of neighboring countries [2,23], up until now there has been no study focused on morphometric and phytochemical data analysis.

Plant *chemophenetics* [24] is a term that was recently proposed for exploring characteristic arrangements of specialized plant taxon metabolites; this analysis contributes to the phenetic description of the taxa—similar to anatomical, morphological, and karyological approaches—and represents an opportunity to describe organisms classified with molecular methods. Thus, the specialized metabolism products could be treated as phenotypic characters that can be used as arguments, e.g., the existence of botanical varieties in the same way as, e.g., traditional morphological characters [25].

Hence, the present study aims to apply a chemophenetic [24] approach to three *Senecio* (*S. hercynicus, S. ovatus*, and *S. rupestris*) and two *Jacobaea* species (*J. pancicii* and *J. maritima*). The morphometric data and phytochemical profiling of phenolic specialized metabolites are combined to give insight into the similarity of species in the *Senecio* taxa.

## 2. Results and Discussion

### 2.1. Morphometric Analysis

Samples for each of the three Senecio species (*S. hercynicus*, *S. ovatus*, and *S. rupestris*) and two Jacobaea species (*J. pancicii* and *J. maritima*) were characterized by 12 independent variables, (X_1–12_) as follows: X_1_—root diameter [cm]; X_2_—stem height [cm]; X_3_—leaf length [cm]; X_4_—leaf width [cm]; X_5_—involucral bract length [cm]; X_6_—involucral bracts number per capitula; X_7_—ray flower length [cm]; X_8_—number of ray flowers per flower head; X_9_—disc flower length [cm]; X_10_—number of disc flowers per flower head; X_11_—flower head diameter [cm]; and X_12_—number of capitula per plant. These variables have been ordinarily applied to differentiate the studied taxa [20,26,27,28]. The raw morphometric data, together with the descriptive statistics, are presented in Appendix A, respectively. A combination of parametric (MANOVA) and non-parametric (Kruskal–Wallis) tests, together with post-hoc Bonferroni tests, were used to derive relationships between the tested variables and the samples’ taxonomic membership. An α level of 0.05 was set as significant. Some notable relationships will be drawn. Root diameter (X_1_) was the only parameter by which a discrimination was evident between samples belonging to the different genera; however, without differentiation within species of the same genus, i.e., *J. pancicii* was not differentiated from *J. maritia*. Similarly, the three Senecio species showed a relative homogeneity on the X_4_ parameter, and were distinguished from the two Jacoboea species; X_4_ also differentiated *J. pancicii* from *J. maritima*. On the other hand, parameters the X_2_, X_5–7_ showed similarity between *S. rupestris* and *J. maritima*, and between *S. ovatus*, *S. hercynicus*, and *J. pancicii*. The other parameters showed quite different relationships to the species, and it is evident from the LDA analysis shown below that a combination of the parameters is needed for the confident differentiation of the discussed species.

#### Correlation and Linear Discriminant Analysis (LDA)

The correlation matrix (Figure 1) revealed that stem height (X_2_) was positively correlated to leaf length (X_3_) and negatively correlated to the variables X_6_, X_8_, and X_10_. The number of ray flowers (X_8_) was positively associated to the involucral bracts number (X_6_) and number of disc flowers (X_10_). Additionally, the variables X_6_, X_8_, and X_10_ showed a high positive correlation between each other.

Next, a linear discriminant analysis (LDA) was performed on the X_1-12_ variables [29,30]. Based on several calculated parameters (Appendix A), including the residual sum of squares (RSS), adjusted R2, Mallow’s Cp, and Bayesian information criterion (BIC), a six variable model was selected, including the variables X_1_, X_4_, X_7_, X_8_, X_9_, and X_11_ (Appendix A). Then, 80% of the data was used as a training set (*n* = 60) and 20%—as a test set (*n* = 15), on a random principle. The training set was used to derive a linear model for predicting the species of a plant, based on the selected set of six parameters. The linear model was able to correctly predict the species on the test set (*n* = 15), but one (Appendix A). On the one-dimensional plot derived from the linear model (Figure 2A), *S. ovatus* and *S. hercynicus* were not well-distinguished, while on the two-dimensional plot, all species were separated, except *S. ovatus* and *S. hercynicus* with a partial overlap (Figure 2B).

Given these results, the morphological variability of the *Senecio* and *Jacobaea* species is not random and is long-established for the tribe Senecioneae taxa as prominent [31]. Although the *Jacobaea* is distinguished from *Senecio sensu stricto*, a clear morphological synapomorphies for *Jacobaea* have not yet been recognized [1]. The received data by the morphometrical study unequivocally confirm the taxonomical relationship of *S. hercynicus* and *S. ovatus* belonging to *S. nemorensis* group and the transfer of the last-mentioned species to the genus *Senecio*. Moreover, the results favor the delimitation of *J. maritima* and *J. pancicii* from the genus *Senecio* and the distinguishing of the other studied taxa [20].

### 2.2. UHPLC-HRMS Identification and Tentative Annotation of Specialized Natural Products

In order to establish a phenolic metabolite profiling, combined hydromethanolic plant extracts were prepared, i.e., LC-MS measurements on phenolic content were performed on the homogenized samples from the 15 collected plants from each species, as described in Section 3.4. Based on chromatographic retention times, MS and MS/MS accurate measurements, fragmentation patterns, and the comparison with reference standards and literature data, a total of 46 hydroxybenzoic, hydroxycinnamic, and acylquinic acids and their derivatives, 1 coumarin and 21 flavonoids, were annotated in the tested extracts. The LC-MS and MS/MS data of all 68 identified phenolic compounds are presented in Table 1 along with their distribution in the studied extracts.

#### 2.2.1. Hydroxybenzoic, Hydroxycinnamic Acids, and Their Glycosides

Five hydroxybenzoic acids (compounds **3**–**5**, **16** and **42**) and four hydoxycinnamic acids (compounds **14**, **26**, **27**, and **36**) along with *p*-hydroxyphenylacetic acid (compound **24**) were identified in the studied species by comparison with reference standards (Table 1). 

Compounds **1**, **2**, **6**–**8**, **10**, **11**, **15**, **17**, **19**, **22**, **25**, and **28** presented similar fragmentation patterns indicating phenolic acid-hexosides. They gave indicative fragment ions at *m*/*z* 153.018 (compound **1**), *m*/*z* 167.034 (compound **2**), *m*/*z* 151.039 (compound **6**), *m*/*z* 197.045 (compound **8**), *m*/*z* 179.034 (compounds **10**, **19**, **22**, **28**), *m*/*z* 137.023 (compounds **11**, **17**), *m*/*z* 193.050 (compound **15**), *m*/*z* 163.039 (compound **26**), and fragmentation pathways consistent with protocatechuic, vanillic, *p*-hydroxyphenylacetic, syringic, caffeic, 4-hydroxybenzoic, ferulic, and *p*-coumaric acid, respectively (Table 1). 

Compound **33** afforded a base peak at *m*/*z* 151.039 [(M−H)−162-42]^−^, and fragment ions at *m*/*z* 123.008 [(M−H)−162-60]^−^ and 109.028 [(M−H)−162-2×42]^−^, suggesting two acetyl groups and a hexose unit. Thus, **33** was ascribed to acetoxy-hydroxyacetophenone-*O*-hexoside. MS/MS of **34** at *m*/*z* 595.131 [M−H]^−^ was acquired; taraxafolin B residue was deduced from the abundant ions at *m*/*z* 341.0883 [M−H−C_11_H_10_O_7_]^−^ (25.4%) and 253.035 [taraxafolin (TF)−H]^−^(23.5%), supported by *m*/*z* 209.045 [TF−H−CO_2_]^−^, 191.034 [TF−H−H_2_O−CO_2_]^−^ and 165.055 [TF−H−2CO_2_]^−^. Accordingly, **34** was tentatively identified as taraxafolin B-(caffeoyl)-hexoside (Table 1).

#### 2.2.2. Acylquinic Acids

Six mono-, nine di- and one triacylquinic acids (AQAs) were identified or annotated in the assayed species. Fragmentation behaviors were consistent with those reported [32,33]. Thus, **23**, **29**, and **35** were assigned to 4-caffeoyl-, 5-*p*-coumaroyl-, and 5-feruloylquinic acid, respectively. diAQA belongs to four widely spread in Asteraceae subclasses: dicaffeoylquinic acids (diCQA) (compounds **37**–**40**), feruloyl-caffeoylquinic acids (FCQA) (compounds **45**, **46**), *p*-coumaroyl-caffeoylquinic acids (*p*-CoCQA) (compounds **43**, **44**), and 3-hydroxy-dihydroxy-5-caffeoylquinic acid (HC-CQA) (compound **30**). 

Compounds **43** and **45** gave abundant ions at *m*/*z* 337.093 (74%) and 367.104 (99%), respectively, indicating a loss of caffeoyl residue before the *p*-coumaroyl (compound **43**) and feruloyl (compound **45**) moiety. Moreover, both compounds gave base peaks at *m*/*z* 163.039 and 193.050, as observed in 3AQA, accompanied by *m*/*z* 119.049 [*p*-coumaric acid-H−CO_2_]^−^ (34%) (compound **43**) and 134.036 [ferulic acid−H−CH_3_−CO_2_]^−^ (69%) (compound **45**) (Table 1). Thus, **43** and **45** were assigned to 3-*p*-Co-5CQA and 3F-5CQA, respectively. In the same way, **44** and **46** were annotated as 3C-5-*p*-CoQA and 3C-5FQA, witnessed by the base peak at *m*/*z* 191.055 [quinic acid−H]^−^ as was seen in 3CQA. Likewise, the base peak at *m*/*z* 191.055, together with the abundant ions at *m*/*z* 179.034 and 135.044 defined 3,5-diCQA, while 1,5-diCQA was deduced from the low abundant dehydrated ion at *m*/*z* 335.078. Vicinal *di*CQA 3,4-diCQA (compound **37**) and 4,5-diCQA (compound **40**) were witnessed by the distinctive dehydrated ion at *m*/*z* 173.045; this assumption is supported by the chromatographic behavior of both compounds on the reverse phase support being the most polar and lipophilic isomers among the *di*CQA [34]. It was noted that 3,4,5-triCQA (compound **47**) was discernable by the prominent ions at *m*/*z* 179.034, 173.045, and 135.044, as was observed in the 3,4-disubstituted quinic acid skeleton.

#### 2.2.3. Flavonoids

The flavonoid annotation was based on the fragmentation patterns for different flavonoid classes previously reported in a few Asteraceae species [32,33,34], or by using flavonoid standards. Retro-Diels-Alder (RDA) fragmentation allowed for the differentiation of flavon and flavonol derivatives. Thus, quercetin (compounds **51**, **53**, **54**, **57**, and **61**), kaempferol (compounds **49, 56,** and **58**) and isorhamnetin (compounds **50**, **52**, **59**, and **60**) flavonols were identified from the RDA ions ^1,3^B^−^, ^1,3^A^−^, ^0,4^A^−^, ^1,2^A^−^, and ^1,2^B^−^ (Table 1). Compounds **51**, **53**, **56**, **58**, **59**, **61,** and **62**–**66** were unambiguously identified by comparison with reference standards. Compound **48** ([M−H]^−^ at *m*/*z* 595.168) showed a typical fragmentation of *C*-glycosylflavone, yielding a series of fragment ions at *m*/*z* 475.125 [M−H−120]^−^, 415.104 [M−H−120−60]^−^, 385.093 [M−H−120−90]^−^, 355.0822 [M−H−2×120]^−^ [35]. The aglycone naringenin in **48** was evidenced by the RDA ions at *m*/*z* 119.049 (^1,3^B^−^), and 107.012 (^0,4^A^−^). Thus, **48** was annotated as 6,8-di*C*-hexosyl-naringenin. In **49** and **50**, the consecutive loss of two hexose units (2×162 Da) is related to an *O*-dihexosyl linkage, while in **52** *O*-pentosylhexosyl linkage was suggested. Compounds **54**, **55**, and **60** presented similar fragmentation patterns yielding base peaks at *m*/*z* 301.036 (compound **54**), 285.041 (compound **55**), and 315.052 (compound **60**) [M−H−HexA]^−^, respectively, indicating flavonoid hexuronides. In the case of **57**, a loss of an acetyl moiety at *m*/*z* 463.089 allowed to annotate quercetin 3-*O*-acetylhexoside. Unlike the Asteraceae species, only two 6-methoxylated flavonoids, **67** and **68**, were suggested on the base of the transitions: 329.067→314.044→299.020 (**67**) and 313.072→298.048→283.025 (**68**). Accordingly, **67** and **68** were ascribed to cirsiliol and cirsimaritin, respectively (Table 1).

### 2.3. Chemophenetic Significance

The chemophenetic significance of phenolic metabolite profiling coupled to morphometric data of the studied Senecioneae species is presented. The raw LC-MS data of annotated specialized compounds were converted and further manipulated with the R programming language, as detailed in Section 3.6. Integration of the Full-MS intensity signals corresponding to the identified compounds allowed the determination of their AUC values. These AUC values were used as a relative quantitative measure for a particular compound, between the studied species. In order to do so, the AUC values were normalized from 0 to 100. Thence, a similarity/dissimilarity clustering analysis of the species was conducted for those compounds found in at least two, out of all five, species (Table 2, Figure 3 and Figure 4). 

Figure 3 depicts a heatmap of the AUC values from Table 2. The dendrograms separated the compounds (columns) into five clusters, and the species (rows) into two clusters (Figure 3). 

The clustering, by rows, did not differentiate the two genera. The greatest resemblance was generated between *S. ovatus* and *S. hercynicus*; *J. pancicii* showed greater similarity to the last-mentioned two species compared to *J. maritima*; *S. rupestris* was cast as a separate node. Appendix A presents the grouped compounds from Figure 3, where it is notable which compounds were characteristic for a given species. Hence, the contribution of the annotated phenolic compounds to the phenetic description of the selected taxa was determined. For example, hydroxybenzoic (**1**, **4**, **6**, **11**, **16**, **17**, and **42**) and hydroxycinnamic (**10**, **15**, **19**, **22**, **27**, and **28**), derivatives as well as the flavonol glucosides (**57** and **58**) were dominant in *S. rupestris*, while diAQAs (**37**, **38**, **39**, **40**, **43**, **44**, and **46**), triAQA (**47**) and the flavonols (**53**, **59**, **63**, and **65**) were in the highest amount in *J. maritima*. The coumarin **12**, acylquinic acids (**35** and **45)** and flavonoid hexuronides (**54**, **55**, and **60**) were characteristic for *S. ovatus. J. pancicii*, on the other hand, presented the highest amount of AQAs (**9**, **23**, and **30**), hydroxycinnamic (**5** and **8**), and flavonol (**48**, **51**, and **56**) derivatives. A heatmap of the Euclidean distance and a PCA plot (of the data in Table 2) are shown in Figure 4, where similar clustering is observed, compared to that in Figure 3.

As phenolic content varies between different plant parts, the %CV of the morphometric characteristics, recorded for each plant species, were typically below 20%CV, except for the number of capitula per plant (X_12_) reaching above 40%CV (Appendix A). Noteworthy, for each of the plant species, the LC-MS measurements on the phenolic content were performed on homogenized samples from all 15 aerial plant samples, providing a representative phenolic profile. Overall, the morphometric data (Figure 2) corroborates the taxonomical relationship of *S. hercynicus* and *S. ovatus* to the *S. nemorensis* group. Moreover, a delimitation was observed between the two *Jacoboea* species (*J. maritima* and *J. pancicii*) from the genus *Senecio* and distinguishing of the other studied taxa [20], and similar findings were detected by to the unsupervised clustering methods applied on the phytochemical data (Figure 3 and Figure 4). In both morphometric characteristics and phenolics content, *S. hercynicus* and *S. ovatus* showed the highest similarity.

## 3. Materials and Methods

### 3.1. Plant Material

The herbal drugs (aerial parts) were collected during the full flowering stage in July 2021, with the location coordinates as follow: *S. hercynicus* at Vitosha Mt., “Zlatni mostove” locality at 1404 m a.s.l. (42.41° N 23.23° E); *S. ovatus*, *S. rupestris*, *J. pancicii* at Vitosha Mt., “Aleko hut” locality at 1855 m a.s.l. (42.58° N 23.29° E); *J. maritima* at the Black Sea coast, “Golden sand” resort at 24 m a.s.l. (43.28° N 28.04° E). The collected taxa at Vitosha Mt. inhabited one and the same plant community. The plant species were identified according to Vladimirov, 2012 [20]. A voucher specimen of *S. hercynicus* was deposited at Herbarium Academiae Scientiarum Bulgariae (SOM 177012). *S. ovatus*, *S. rupestris*, *J. pancicii*, and *J. maritima* specimens were given at Herbarium Facultatis Pharmaceuticae Sophiensis, Medical University-Sofia, Bulgaria (Voucher specimen № 11 631–11 634).

### 3.2. Morphometric Measurements

Morphometric measurements on the studied *Senecio* and *Jacobaea* species were performed on 15 randomly chosen plants, from each species, during the full flowering stage. The morphometric variability was determined using 12 quantitative characters (parameters) as follows: X_1_—root diameter [cm]; X_2_—stem height [cm]; X_3_—leaf length [cm]; X_4_—leaf width [cm]; X_5_—involucral bract length [cm]; X_6_—involucral bracts number per capitula; X_7_—ray flower length [cm]; X_8_—number of ray flowers per flower head; X_9_—disc flower length [cm]; X_10_—number of disc flowers per flower head; X_11_—flower head diameter [cm]; and X_12_—number of capitula per plant. The morphometric measurements are presented in Appendix A. Descriptive statistics of the 12 characteristics was performed in the R programming language and presented in Appendix A.

### 3.3. Chemicals and Reagents

Acetonitrile and formic acid for LC-MS, and methanol of analytical grade, were purchased from Merck (Merck, Bulgaria). The reference standards used for compound identification were bought from Phytolab (Vestenbergsgreuth, Germany).

### 3.4. Sample Extraction and Sample Preparation

Air-dried powdered aerial parts (5 g, combined plant material belonging to the same species) were extracted with 80% MeOH (1:20 *w*/*v*) by sonication (100 kHz) for 15 min (×2) at room temperature. Then, the extracts were concentrated *in vacuo* and lyophilized to yield crude extracts: *S. hercynicus*—0.74 g, *S. ovatus*—0.71 g, *S. rupestris*—1.02 g, *J. pancicii*—0.95 g, and *J. maritima*—0.96 g.

### 3.5. Ultra-High-Performance Liquid Chromatography—High Resolution Mass Spectrometry (UHPLC-HRMS)

Elution was achieved on a reversed phase column Kromasil EternityXT C18 (1.8 µm, 2.1 × 100 mm, AkzoNobel, Sweden) column maintained at 40 °C. The binary mobile phase consisted of A: 0.1% formic acid in water and B: 0.1% formic acid in acetonitrile. The run time was 24.5 min. Prior to injection, the mobile phase was held at 50% B for 4.5 min, and then gradually turned at 5% B in 0.5 min. After injection, the % B was gradually turned to 60% B over 15 min, and then held at 60% B for 3 min, increased gradually to 95% B over 3 min, held at 95% B over 2 min, then turned to 50% B in 0.5 min. The retention time of the identified compounds ranged between 1.74 and 9.60 min. The flow rate and the injection volume were set to 300 µL/min and 1 µL, respectively. The effluents were connected on-line with a Q Exactive Plus Orbitrap mass spectrometer (ThermoFisher Scientific) where the compounds were detected. Data were processed with Xcalibur software 4.2 (ThermoFisher Scientific, Waltham, MO, USA). 

Mass spectrometric analyses were carried out on a Q Exactive Plus Mass Spectrometer (ThermoFisher Scientific) equipped with a heated electrospray ionization (HESI-II) probe (ThermoFisher Scientific). The tune parameters were as follows: spray voltage 3.5 kV; sheath gas flow rate 38; auxiliary gas flow rate 12; spare gas flow rate 0; capillary temperature 320 °C; probe heater temperature 320 °C, and S-lens RF level 50. Acquisition was acquired at Full-scan MS and Data Dependent-MS^2^ modes. Full-scan spectra over the *m*/*z* range 100 to 1000 were acquired in the negative ionization mode at a resolution of 70,000. Other instrument parameters for the Full MS mode were set as follows: AGC target 1e6, maximum ion time 80 ms, number of scan ranges 1. For DD-MS^2^ mode, instrument parameters were as follows: microscans 1, resolution 17,500, AGC target 1e5, maximum ion time 50ms, MSX count 1, isolation window 1.0 *m*/*z*, stepped collision energy (NCE) 10, 30, and 60. Data acquisition and processing were carried out with Xcalibur 4.2 software (ThermoFisher Scientific).

### 3.6. File Conversions and Data Analysis

After the .raw (ThermoFisher Scientific) mass spectrometric files were obtained, they were converted to .ms1 (MS1 data) and .mgf (MS2 data) files using MSConvertGUI 3.1 (ProteoWizard). Then, the .ms1 and .mgf files were imported to RStudio (2021, Build 382) and further manipulated under the R programming language (version 4.2.1, 23 June 2022, “Funny-Looking Kid”). The MS2 spectra were screened for the presence of the available target (hydroxybenzoic acid derivatives and flavonoids) standard compounds. The screening was achieved by selecting spectra based on the following criteria: *m*/*z* error of the molecular ion < 15 ppm (minimum 0.0010 Da), retention time error < 2% (minimum 0.05 min, maximum 0.15 min), number of fragment ions match > 2/3, and error of the percentage intensity of matched fragment ion < 15. Similar MS2 scans found in the same chromatographic peak were grouped, i.e., the spectra were summed, the *m*/*z* were adjusted by weight averaging:$${\left(m/z\right)}_{avg}=\frac{\sum_{i=1}^{N}int_{i}\times {\left(m/z\right)}_{i}}{N}\;$$
where ${\left(m/z\right)}_{avg}$ is the recalculated *m*/*z* value, ${\left(m/z\right)}_{i}$ and $int_{i}$ are the *m*/*z* and the intensity of the *i*th fragment ion, respectively. The areas under the curve (AUC) of the identified compounds were calculated and normalized from 0 to 100. 

Data analysis was performed in the R programming language (R version 4.2.1., 23 June 2022, Funny-Looking Kid), operated under the RStudio environment (2022.07.2 Build 576). R packages used include: “MASS” [36], “klaR” [37], “caret” [38], “leaps” [39], “factoextra” [40], “cluster” [41], “lpSolve” [42], “DescTools” [43], “pheatmap” [44], and “arsenal” [45]. Distance matrices were generated using the “Euclidean” method, and hierarchical clustering was performed using the “ward.D2” method. The complete R code used for morphometric analysis is presented in the Appendix A.

## 4. Conclusions

Herein, a chemophenetic study of three *Senecio* (*S. hercynicus, S. ovatus*, and *S. rupestris*) and two *Jacobaea* species (*J. pancicii* and *J. maritima*) is presented. From the collected morphometric data, describing 12 parameters, a distinguishment of species by genera was performed using linear discriminant analysis (LDA). Among the studied species, *S. hercynicus* and *S. ovatus* presented the greatest similarity, and hence, their formed clusters were the closest. Even though no overlap in the LDA analysis was observed between the Jacoboea and Senecio species, *J. pancicii* and *J. maritima* did not demonstrate likeness. A phytochemical analysis by UHPLC-Orbitrap-HRMS revealed a total of 46 hydroxybenzoic, hydroxycinnamic, and acylquinic acids and their derivatives, 1 coumarin and 21 flavonoids. Hierarchical and PCA clustering was then applied to the phytochemical data on combined plant material from each species. The data corroborated the similarity of *S. hercynicus* and *S. ovatus,* established in the morphometric analysis. The study highlights the similarity/dissimilarity, both morphometric, and in a manner of specialized metabolites, of the selected species belonging to *Senecio* and *Jacobaea* genera (Senecioneae).

## Figures and Tables

**Figure 1 plants-12-00390-f001:**
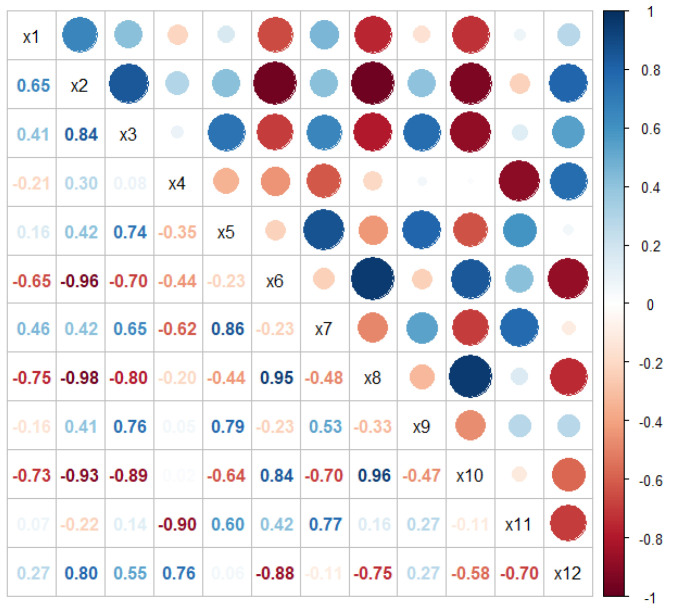
Correlogram of the 12 morphometric characters.

**Figure 2 plants-12-00390-f002:**
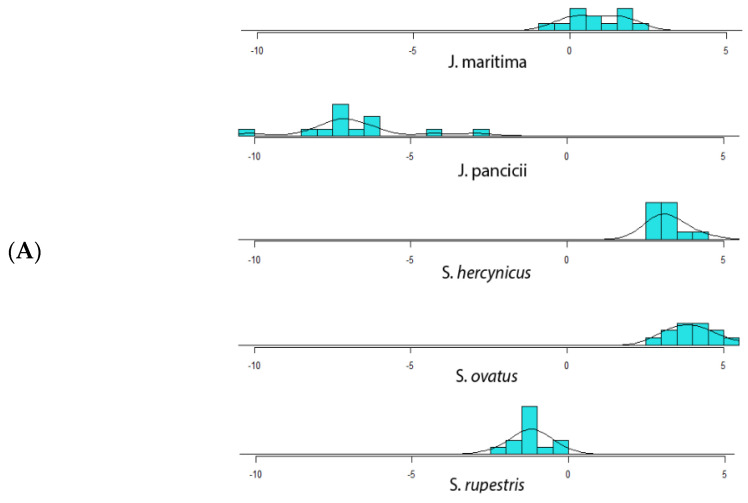
Discriminatory power of LD1 and LD2 functions. (**A**)—one-dimensional (1D) and (**B**)—two-dimensional (2D) discrimination.

**Figure 3 plants-12-00390-f003:**
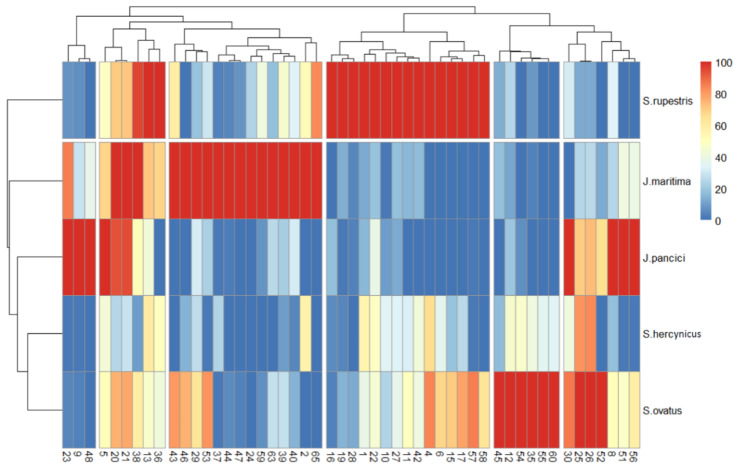
Heatmap of the normalized AUC values, from 0 (in case the compound was not detected) to 100, of the identified compounds (columns) by species (rows).

**Figure 4 plants-12-00390-f004:**
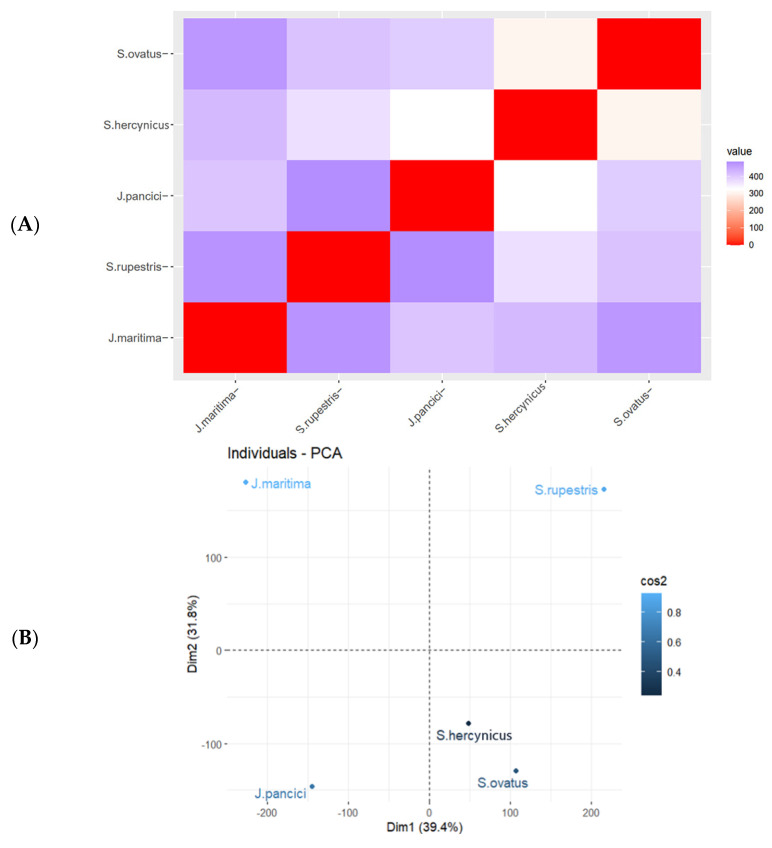
(**A**): Heatmap of the Euclidean distance using the normalized (0 to 100) AUC of identified compounds. In case a compound was not detected in an extract, the AUC for that compound was set to 0; (**B**): Principal component analysis (PCA) (cos2—quality of representation).

**Table 1 plants-12-00390-t001:** Specialized metabolites identified by UHPLC-HRMS. From a total of 68 annotated substances, 24 were identified by a reference standard (marked with *).

№	Annotated Compounds	Molecular Formula	Exact Mass [M−H]^−^	MS^2^	t_R_ (Min)	Distribution
**Hydroxybenzoic, Hydroxycinnamic and Acylquinic Acids, Their Derivatives and Coumarin**						
1	protocatechuic acid-*O*-hexoside	C_13_H_16_O_9_	315.0722	315.0726 (100), 153.0182 (29.5), 152.0104 (60.9), 109.0284 (10.1)	1.74	A, B, C, D, E
2	vanillic acid 4-*O*-hexoside	C_14_H_18_O_9_	329.0878	329.0885 (1.8), 167.034 (100), 152.0103 (23), 123.0438 (14.3), 108.0202 (37.8)	1.81	A, B, C, E
3	syringic acid *	C_9_H_10_O_5_	197.0456	197.0449 (16.5), 182.0211 (3.2), 153.0549 (8.9), 138.0314 (3.3), 123.0437 (58.3)	1.76	C
4	vanillic acid *	C_8_H_8_O_4_	167.0350	167.0339 (31.1), 152.0103 (100), 123.0438 (32.1), 108.0202 (52.9), 95.0486 (8)	1.82	A, B, C, D, E
5	protocatechuic acid *	C_7_H_6_O_4_	153.0193	153.0182 (15.2), 109.0281 (100), 91.0173 (1.2), 81.033 (1.4)	2.04	A, B, C, D, E
6	*p*-hydroxyphenylacetic acid-*O*-hexoside	C_14_H_18_O_8_	313.0929	313.0923 (13), 151.0389 (100), 133.0284 (0.2), 123.0075 (0.8), 109.0281 (4.2)	3.00	A, B, C
7	gluconic acid-*O*-hexoside	C_15_H_18_O_10_	357.0827	357.083 (100), 195.0293 (10.8), 177.0183 (8), 151.039 (71.8)	2.25	C
8	syringic acid 4-*O*-hexoside	C_15_H_20_O_10_	359.0984	359.0985 (8), 197.0448 (100), 182.0212 (21.7), 153.0546 (16.1), 138.031 (29.3), 123.0074 (33.1)	2.27	A, B, C, D, E
9	neochlorogenic acid *	C_16_H_18_O_9_	353.0878	353.0882 (46.1), 191.0553 (100), 179.0341 (68.1), 173.0444 (4.1), 161.0235 (5.9), 135.0439 (54.4), 127.0385 (0.9), 111.0438 (0.7), 93.0329 (3.7), 85.0279 (8.5)	2.37	A, B, C, D, E
10	caffeic acid-*O*-hexoside	C_15_H_18_O_9_	341.0878	341.0884 (2.2), 179.034 (2.9), 135.0438 (100), 107.0488 (0.7)	2.54	A, B, C, D, E
11	4-hydroxybenzoic acid-*O*-hexoside	C_13_H_16_O_8_	299.0773	299.0775 (13.7), 137.0231 (100), 93.033 (0.2)	2.46	A, B, C, E
12	esculetin-*O*-hexoside	C_15_H_16_O_9_	339.0722	339.0721 (10.6), 177.0184 (100), 149.0233 (0.9), 133.0282 (8), 105.0331 (3.3), 89.0381 (2.4)	2.72	A, B, C, D, E
13	4-hydroxybenzoic acid	C_7_H_6_O_3_	137.0244	137.0232 (100), 108.0203 (11.2), 93.0333 (3.3)	2.84	A, B, C, D, E
14	ferulic acid *	C_10_H_10_O_4_	193.0506	193.05 (100), 178.0264 (74.8), 163.0391 (34.4), 149.0598 (38), 134.036 (82.5)	2.96	A, B, C
15	ferulic acid-*O*-hexoside	C_16_H_20_O_9_	355.1035	355.1048 (1), 193.0499 (100), 178.0263 (10.9), 149.0596 (21.4), 134.036 (62.1)	2.96	A, B, C
16	gentisic acid *	C_7_H_6_O_4_	153.0193	153.0182 (46.7), 123.0074 (20.8), 109.0283 (40.6), 81.0331 (5.4)	2.98	A, B, C, D
17	4-hydroxybenzoic acid-*O*-hexoside isomer	C_13_H_16_O_8_	299.0773	299.0783 (1.3), 137.0231 (100), 93.033 (50.8)	3.00	A, B, C
18	3-*p*-coumaroylquinic acid	C_16_H_18_O_8_	337.1500	191.0554 (19.6), 163.039 (100), 161.0443 (4.2), 119.0488 (23.7)	3.04	C
19	caffeic acid-*O*-hexoside	C_15_H_18_O_9_	341.0878	341.088 (26.8), 179.034 (100), 135.0438 (77), 107.0486 (0.8)	3.07	A, B, C, D, E
20	quinic acid	C_7_H_12_O_6_	191.0561	191.0553 (100), 173.0446 (2), 155.0338 (0.2), 127.0388 (4.3), 111.0437 (1.9), 93.0331 (6.4), 85.0279 (18.1)	3.19	A, B, C, D, E
21	chlorogenic acid	C_16_H_18_O_9_	353.0878	353.0881 (3.9), 191.0553 (100), 179.0343 (1.1), 173.0449 (0.4), 161.0232 (1.6), 135.0439 (0.5), 127.0386 (1.3), 111.0433 (0.3), 93.033 (2.2), 85.0279 (7.2)	3.19	A, B, C, D, E
22	caffeic acid-*O*-hexoside isomer I	C_15_H_18_O_9_	341.0878	341.0881 (9.5), 179.034 (100), 135.0438 (60.8), 107.049 (0.6)	3.27	A, B, C, D, E
23	4-caffeoylquinic acid	C_16_H_18_O_9_	353.0878	353.0882 (32.1), 191.0554 (97.5), 179.0341 (72.6), 173.0446 (100), 135.0439 (56.3), 127.0387 (1.8), 111.0435 (3.3), 93.0331 (22), 85.028 (11.3)	3.37	A, B, C, D, E
24	*p*-hydroxyphenylacetic acid	C_8_H_8_O_3_	151.0401	151.0389 (100), 136.0154 (2), 123.0074 (4.2), 109.028 (15)	3.47	C, E
25	coumaric acid-*O*-hexoside	C_15_H_18_O_8_	325.0929	325.0923 (1.7), 163.039 (100), 145.0284 (3.5), 119.0488 (92.1), 93.0333 (0.8)	3.33	A, B, C, D, E
26	*p*-coumaric acid *	C_9_H_8_O_3_	163.0401	163.0389 (6.7), 135.0438 (0.7), 119.0488 (100)	3.33	A, B, C, D, E
27	caffeic acid *	C_9_H_8_O_4_	179.0350	179.0341 (20.5), 135.0438 (100), 117.0332 (0.7), 107.0489 (1.4)	3.53	A, B, C, D, E
28	caffeic acid-*O*-hexoside isomer II	C_15_H_18_O_9_	341.0878	341.088 (24.5), 179.0341 (100), 135.0439 (85.6), 107.0489 (0.5)	3.79	B, C, D, E
29	5-*p*-coumaroylquinic acid	C_16_H_18_O_8_	337.0929	337.0933 (8.7), 191.0554 (100), 173.0449 (6), 163.0389 (5.7), 127.0391 (1), 119.0489 (4.8), 111.0437 (1.9), 93.033 (17.2), 85.028 (4.9)	3.95	A, B, C, D, E
30	3-hydroxy-dihydrocaffeoyl-5-caffeoylquinic acid	C_25_H_26_O_13_	533.1301	533.1306 (100), 191.0554 (83.4), 173.0447 (10.1), 161.0596 (3.2), 127.0387 (3.3), 93.033 (17.8), 85.028 (11.8)	3.09	A, B, C, D, E
31	isoferulic acid	C_10_H_10_O_4_	193.0506	193.0499 (100), 178.0265 (0.8), 163.0391 (41.6), 149.0597 (18.8), 135.0439 (38.8)	4.10	C
32	syiringaldehide	C_9_H_10_O_4_	181.0506	181.0497 (15.2), 166.0261 (100), 151.0025 (58.4), 123.0074 (15.7)	4.22	C
33	acetoxy-hydroxyacetophenone-*O*-hexoside	C_16_H_20_O_9_	355.1035	355.1039 (13.7), 193.0494 (1.2), 151.0389 (100), 123.0076 (0.9), 109.0281 (4.4)	4.27	C
34	taraxafolin B-(caffeoyl)-hexoside	C_26_H_28_O_16_	595.1305	595.1308 (100), 341.0883 (25.4), 253.0353 (23.5), 235.0245 (3.5), 209.0446 (1.4), 191.0341 (31.1), 179.0341 (93.7), 165.0545 (16.4), 135.0438 (56)	4.41	C
35	5-feruoylquinic acid	C_17_H_20_O_9_	367.1035	367.1038 (15.4), 191.0554 (100), 173.0445 (10.5), 134.0359 (13.1), 111.0437 (3.8), 93.0331 (25.5), 85.028 (5.1)	4.42	A, B, C, E
36	*m*-coumaric acid *	C_9_H_8_O_3_	163.0401	163.039 (7.6), 135.0439 (0.5), 119.0489 (100)	4.56	A, B, C, E
37	3,4-dicaffeoylquinic acid *	C_25_H_24_O_12_	515.1195	515.1198 (100), 353.0881 (15), 335.0773 (5.3), 203.0344 (0.5), 191.0555 (29.1), 179.0341 (53), 173.0446 (58.7), 161.0233 (17), 135.0439 (53.1), 127.0386 (2.2), 111.0437 (3.6), 93.0331 (16.8), 85.028 (3.7)	5.69	A, B, C, D, E
38	3,5-dicaffeoylquinic acid *	C_25_H_24_O_12_	515.1195	515.1202 (13.5), 353.0881 (100), 191.0554 (91.3), 179.0341 (49.4), 173.0443 (3.7), 161.0234 (4.2), 135.0439 (55.8), 111.0437 (1.3), 93.0332 (4.2), 85.028 (9.8)	5.86	A, B, C, D, E
39	1,5-dicaffeoylquinic acid *	C_25_H_24_O_12_	515.1195	515.1199 (25.5), 353.088 (92.4), 335.0777 (1.9), 191.0554 (100), 179.0341 (53.3), 173.0446 (8.7), 135.0439 (65), 127.0387 (4.2), 111.0437 (2.1), 93.0332 (6.5), 85.028 (10.2)	6.02	A, B, C, D, E
40	4,5-dicaffeoylquinic acid *	C_25_H_24_O_12_	515.1195	515.1197 (100), 353.0883 (72.3), 203.0341 (1.5), 191.0553 (38.9), 179.0341 (66.6), 173.0446 (98.1), 135.0439 (69.5), 111.0435 (5.2), 93.033 (30.8), 85.0279 (8.3)	6.22	A, B, C, D, E
41	shikimic acid	C_7_H_10_O_5_	173.0456	173.0444 (100), 111.0437 (10), 93.033 (68.4)	6.22	E
42	salicilic acid *	C_7_H_6_O_3_	137.0244	137.023 (8.7), 93.0331 (100)	6.29	A, C
43	3-*p*-coumaroyl-5-caffeoylquinic acid	C_25_H_24_O_11_	499.1246	499.1238 (16.4), 353.0901 (1.5), 337.0933 (73.9), 335.0797 (1.7), 191.0553 (12.4), 173.0449 (7.9), 163.039 (100), 135.0441 (4.2), 119.0489 (34.4), 93.0334 (4.4)	6.51	B, C, E
44	3-caffeoyl-5-*p*-coumaroylquinic acid	C_25_H_24_O_11_	499.1246	499.125 (26.1), 353.0882 (64.8), 337.0938 (17.5), 191.0554 (100), 179.0341 (34.5), 173.0446 (6.9), 163.0389 (2.9), 161.0231 (5.5), 135.0439 (36.8), 119.0488 (2.8), 111.0436 (1.4), 93.0331 (10.5), 85.0279 (7.1)	6.57	B, E
45	3-feruoyl-5-caffeoylquinic acid	C_26_H_26_O_12_	529.1352	529.1296 (2.3), 367.1036 (99.2), 335.078 (1.1), 193.0499 (100), 191.0557 (3), 173.0443 (6.9), 161.0235 (2), 134.036 (68.5), 93.0331 (3.2)	6.82	A, B, C, E
46	3-caffeoyl-5-feruoylquinic acid	C_26_H_26_O_12_	529.1352	529.1353 (41.1), 367.1037 (0.7), 353.0882 (43.5), 335.0794 (0.8), 191.0555 (100), 179.0342 (40.7), 173.0446 (11.5), 161.0238 (5.4), 135.0439 (37.7), 134.0361 (12.9), 127.0383 (1.3), 111.0437 (1.3), 93.0331 (15.5), 85.028 (7.5)	6.89	A, B, C, E
47	3,4,5-tricaffeoylquinic acid	C_34_H_30_O_15_	677.1512	677.1517 (100), 515.1202 (46.2), 353.0883 (47.1), 335.0783 (13.9), 191.0554 (45.2), 179.0342 (65.2), 173.0446 (90.2), 161.0234 (24.1), 135.0439 (72.1), 111.0436 (5.6), 93.0331 (21.7)	7.77	B, C, E
**Flavonoids**						
48	6,8-di-*C*-hexosyl-naringenin	C_27_H_32_O_15_	595.1669	595.1677 (100), 475.1247 (3.8), 457.1138 (1.3), 427.1055 (1.2), 415.1037 (11.6), 385.0933 (36.1), 355.0822 (38.9), 343.0825 (3.9), 313.0722 (6.1), 119.0489 (15.5), 107.0123 (3.5),	3.63	D, E
49	kaempferol-*O*-dihexoside	C_27_H_30_O_16_	609.1461	609.1462 (100), 447.0931 (24.7), 285.0405 (50.3), 284.0325 (7.6), 255.0300 (33.4), 227.0347 (5.7), 211.0391 (2.2)	3.81	C
50	isorhamnetin-*O*-dihexoside	C_28_H_32_O_17_	639.1567	639.1575 (100), 477.1039 (34.6), 315.0514 (56.7), 300.0275 (11.8), 314.0429 (12.6), 285.0408 (6.4), 270.0172 (20.8), 242.0218 (14.0), 227.0344 (0.7), 151.0027 (5.5), 107.0124 (1.3)	4.04	C
51	rutin *	C_27_H_30_O_16_	609.1461	609.1469 (100), 301.0352 (39.6), 300.0279 (64.0), 271.0249 (29.6), 255.0298 (14.6), 243.0296 (6.4), 227.0345 (2.2), 178.9975 (3.4), 163.0015 (0.8), 151.0025 (5.6), 121.0286 (0.9), 107.0125 (1.0)	5.07	A, B, C, D, E
52	isorhamnetin-*O*-pentosylhexoside	C_27_H_30_O_16_	609.1461	609.1464 (100), 315.0504 (19.9), 314.0436 (85.1), 299.0196 (18.4), 271.0252 (22.0), 243.0297 (20.1), 227.0350 (5.1), 178.9978 (1.3), 151.0023 (34.0)	5.16	A, B, C, D, E
53	isoquercitrin *	C_21_H_20_O_12_	463.0882	463.0887 (100), 301.0352 (44.4), 300.0278 (69.8), 271.0250 (46.2), 255.0300 (20.0), 243.0298 (11.0), 227.0346 (4.6), 211.0389 (1.4), 178.9976 (4.4), 151.0026 (7.8), 121.0280 (1.5), 107.0123 (2.8)	5.27	A, B, C, D, E
54	quercetin 7-*O*-hexuronide	C_21_H_18_O_13_	477.0675	477.0671 (47.4), 301.0356 (100), 283.0245 (2.1), 255.0302 (3.1), 227.0343 (2.0), 211.0396 (1.8), 178.9976 (9.9), 163.0028 (2.4), 151.0025 (20.2), 121.0281 (6.3), 107.0124 (8.9)	5.22	A, B, C, D
55	luteolin-*O*-hexuronide	C_21_H_18_O_12_	461.0726	461.0730 (39.5), 285.0406 (100), 257.0457 (4.6), 229.0505 (6.0), 213.0544 (2.0), 175.0242 (5.8), 151.0023 (0.9), 107.0125 (2.5)	5.84	A, B
56	kaempferol 7-*O*-rutinoside *	C_27_H_30_O_15_	593.1512	593.1516 (100), 285.0405 (90.5), 255.0298 (42.9), 227.0346 (31.3), 163.0025 (1.1)	5.62	A, B, D, E
57	quercetin 3-*O*-acetylhexoside	C_23_H_22_O_13_	505.0988	505.0996 (100), 463.0891 (0.8), 301.0351 (34.1), 300.0278 (88.9), 271.0251 (42.8), 255.0299 (21.5), 243.0297 (11.7), 227.0343 (3.1), 178.9976 (2.5), 163.0027 (2.8), 151.0024 (7.8), 121.0283 (1.1), 107.0124 (2.4)	5.61	A, B, C, D
58	kaempferol-3-*O*-glucoside *	C_21_H_20_O_11_	447.0933	447.0938 (100), 285.0401 (21.4), 284.0329 (51.3), 255.0300(38.4), 227.0347 (40.6), 151.0024 (2.7),	5.87	B, C
59	isorhamnetin 3-*O*-glucoside *	C_22_H_22_O_12_	477.1039	477.1041 (100), 315.0495 (9.7), 314.0437 (51.2), 299.0213 (3.2), 271.0251 (18.8), 257.0460 (3.9), 243.0299 (22.3), 227.0341 (2.9), 215.0340 (3.7), 178.9972 (0.6), 151.0021 (1.7)	6.02	A, B, C, D, E
60	isorhamnetin-*O*-hexuronide	C_22_H_20_O_13_	491.0831	491.0836 (48.3), 315.0515 (100), 300.0278 (29.1), 271.0251 24.7), 255.0299 (10.7), 227.0347 (1.8), 175.0238 (5.7), 151.0029 (2.6), 107.0122 (0.8)	6.09	A, B
61	quercetin-3-*O*-rhamnoside (quercitrin) *	C_21_H_20_O_11_	447.0933	447.0936 (100), 301.0355 (81.9), 300.0278 (22.24), 271.0248 (1.7), 255.0298 (2.0), 227.0352 (1.6), 178.9974 (2.6), 151.0025 (40.6), 121.0281 (8.9), 107.0124 (15.2)	6.76	B
62	luteolin *	C_15_H_10_O_6_	285.0405	285.0406 (100), 175.0392 (3.0), 151.0024 (4.7), 133.0282 (22.8), 107.0124 (3.7)	7.56	C
63	Quercetin *	C_15_H_10_O_7_	301.0354	301.0356 (100), 273.0411 (3.3), 257.0469 (1.8), 245.0444 (0.8), 229.0500 (0.6), 215.1699 (0.3), 178.9977 (21.3), 151.0024 (49.4), 121.0281 (14.2), 107.0123 (12.9)	7.61	B, C, D, E
64	apigenin *	C_15_H_10_O_5_	269.0456	269.0458 (100), 225.0549(1.9), 201.0550 (0.9), 151.0025 (5.7), 121.0282 (1.3), 117.0332 (18.4), 107.0124 (5.3)	8.62	C
65	kaempferol *	C_15_H_10_O_6_	285.0405	285.0406 (100), 257.0465 (0.8), 243.0298 (0.2), 227.0353 (0.9), 211.0397 (1.3), 151.0025 (1.3), 107.0123 (1.2)	8.85	C, E
66	chrysoeriol *	C_16_H_12_O_6_	299.0561	299.0564 (63.8), 284.0329 (100), 255.0300 (46.5), 227.0344 (38.2), 211.0394 (1.5), 151.0024 (0.3)	9.32	C
67	cirsiliol	C_17_H_14_O_7_	329.0667	329.0672 (100), 314.0441 (43.6), 299.0199(89.6), 271.0248 (55.9), 243.0296 (6.5), 227.0345 (3.4), 211.1333 (6.8)	9.60	C
68	cirsimaritin	C_17_H_14_O_6_	313.0718	313.0721 (100), 298.0483 (65.2), 283.0251 (52.5), 255.0298 (64.1), 227.0341 (4.4), 211.0396 (5.7)	12.27	C

* identified by reference standards. A—*S. hercynicus*; B—*S. ovatus*; C—*S. rupestris*; D—*J. pancicii*, E—*J. maritima*.

**Table 2 plants-12-00390-t002:** Normalized (by rows) AUC values of the identified specialized natural compounds found in at least two, out of all five, studied species. The cells show the normalized AUC, from 0 (in case the compound was not detected in the extract) to 100, by rows.

№	Compounds	Species				
		*J.maritima*	*J.pancici*	*S.hercynicus*	*S.ovatus*	*S.rupestris*
1	protocatechuic acid-*O*-hexoside	12.64	15.27	55.32	38.45	100
2	vanillic acid 4-*O*-hexoside	100	0	56.87	0	52.35
4	vanillic acid	0	0	66.47	84.99	100
5	protocatechuic acid	68.03	100	43.93	51.5	48.22
6	*p*-hydroxyphenylacetic acid-*O*-hexoside	0	0	36.05	68.05	100
8	syringic acid 4-*O*-hexoside	24.14	100	17.32	51.69	33.42
9	neochlorogenic acid	28.06	100	1.42	3.21	6.19
10	caffeic acid-*O*-hexoside	1.86	9.82	34.24	23.69	100
11	4-hydroxybenzoic acid-*O*-hexoside	15.24	0	32.1	50.1	100
12	esculetin-*O*-hexoside	12	19.44	45.48	100	24.71
13	4-hydroxybenzoic acid	71.02	42.4	60.25	46.55	100
15	ferulic acid-*O*-hexoside	0	0	18.03	70.66	100
16	gentisic acid	0	17.77	5.99	3.02	100
17	4-hydroxybenzoic acid-*O*-hexoside isomer	0	0	25.97	78.36	100
19	caffeic acid-*O*-hexoside	13.08	1.24	4.61	14.54	100
20	quinic acid	100	93.53	24.24	75.9	70.84
21	chlorogenic acida	100	94.47	28.47	77.32	71.97
22	caffeic acid-*O*-hexoside isomer I	18.62	38.53	48.92	44.77	100
23	4-caffeoylquinic acid	85.4	100	1.78	4.3	7.54
24	*p*-hydroxyphenylacetic acid	100	0	0	0	23.6
25	coumaric acid-*O*-hexoside	24.08	70.79	81.31	100	12.01
26	*p*-coumaric acid	25.08	73.49	83.38	100	12.42
27	caffeic acid	17.16	12.38	33.55	37.96	100
28	caffeic acid-*O*-hexoside isomer II	4.28	3.58	0	12.49	100
29	5-*p*-coumaroylquinic acid	100	31.1	26.13	63.12	18.11
30	3-hydroxy-dihydroxy-5-caffeoylquinic acid	0	100	40.05	86.58	31.91
35	5-feruoylquinic acid	3.74	0	40.78	100	8.11
36	*m*-coumaric acid	68.8	0	49.92	41.24	100
37	3,4-dicaffeoylquinic acid	100	1.7	23.32	1.57	3.38
38	3,5-dicaffeoylquinic acid	100	51.54	9.21	59.82	95.36
39	1,5-dicaffeoylquinic acid	100	21.84	9.53	27.34	45.34
40	4,5-dicaffeoylquinic acid	100	31.87	4.14	14.58	33.42
42	salicilic acid	16.54	0	42.11	40.7	100
43	3-*p*-coumaroyl-5-caffeoylquinic acid	100	0	0	80.53	59.01
44	3-caffeoyl-5-*p*-coumaroylquinic acid	100	2.33	0	5.89	4.4
45	3-feruoyl-5-caffeoylquinic acid	17.99	0	16.99	100	13.26
46	3-caffeoyl-5-feruoylquinic acid	100	0	14.92	75.24	0
47	3,4,5-tricaffeoylquinic acid	100	0	0	3.95	7.03
48	6, 8-di-*C*-hexosyl-naringenin	37.7	100	0	0	0
51	rutin	40.39	100	1.62	47.2	0.13
52	isorhamnetin-*O*-pentosylhexoside	12.03	65.54	0	100	0.64
53	isoquercitrin	100	22.06	1.19	81.87	29.95
54	quercetin 7-*O*-hexuronide	0	6.91	46.7	100	0.98
55	luteolin-*O*-hexuronide	0	0	35.07	100	0
56	kaempferol 7-*O*-rutinoside	39.27	100	1.89	58.25	0
57	quercetin 3-*O*-acetylhexoside	0	1.07	1.66	85.28	100
58	kaempferol-3-*O*-glucoside	0	0	0	63.53	100
59	isorhamnetin 3-*O*-glucoside	100	7.5	0.73	6.28	40.77
60	isorhamnetin-*O*-hexuronide	0	0	34.34	100	0
63	quercetin	100	25.2	0	27.19	17.63
65	kaempferol	100	0	0	0	83.75

## Data Availability

The data presented in this study are available in the article or Appendix A. The raw MS files are available on request from the corresponding author.

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
