# Peer review of "Chemophenetic Approach to Selected Senecioneae Species, Combining Morphometric and UHPLC-HRMS Analyses"

_plants, 2023, doi:10.3390/plants12020390_

Round 1
Reviewer 1 Report
The manuscript is interesting, among others, for two reasons: it skillfully combines morphometric analysis with chemophenic study, and, it introduces multivariate statistical techniques (PLS, HCA, PCA) to analyze secondary metabolite information in order to search for similarity/dissimilarity between various plant species. Prior to publication, authors should address the following issues.
The morphometric analysis of the studied Senecio and Jacobaea species was carried out using twelve independent variables (X1 – X12). Authors should specify at the beginning of Section 2. Results and Discussion (lines 68-69) what plant morphological characteristics characterize these variables, notwithstanding that they are explained in Section 3.2. Morphometric measurements. This will significantly improve the readability of the manuscript, for example the readability of Section 2.1.1.
Similarity/dissimilarity clustering analysis was performed using the normalized area under the curve (AUC) of those reference compounds found in at least two of the five species. The authors should explain in detail why they use AUC instead of data on the actual content of phenolic metabolites in the studied plants. AUC only shows the area under the peak, nothing else. Calibration curves should be developed to obtain quantitative data on phenolic metabolites. In addition, Table S5 should be transferred from the Supplementary Materials to the original manuscript, it contains very important information.
Figure 3, Figure 4 (A and B), “S. hercynicus” instead of “S. herc”.
Author Response
Dear reviewer, your efforts into refining the quality of the manuscript are well appreciated!
Comment:
"The morphometric analysis of the studied Senecio and Jacobaea species was carried out using twelve independent variables (X1 – X12). Authors should specify at the beginning of Section 2. Results and Discussion (lines 68-69) what plant morphological characteristics characterize these variables, notwithstanding that they are explained in Section 3.2. Morphometric measurements. This will significantly improve the readability of the manuscript, for example the readability of Section 2.1.1."
Answer:
The twelve independent variables (X1-12) are added at the beginning Section 2. Results and Discussion.
Comment:
Similarity/dissimilarity clustering analysis was performed using the normalized area under the curve (AUC) of those reference compounds found in at least two of the five species. The authors should explain in detail why they use AUC instead of data on the actual content of phenolic metabolites in the studied plants. AUC only shows the area under the peak, nothing else. Calibration curves should be developed to obtain quantitative data on phenolic metabolites.
Answer:
We have reflected on this in the revised manuscript in Section 2.3. Chemophenetic significance.
Comment:
In addition, Table S5 should be transferred from the Supplementary Materials to the original manuscript, it contains very important information.
Answer:
Dear reviewer, Table S5 representing the normalized AUC values was placed in the supplementary, because this data is more attractively presented in Figure 3, where the colors of the cells represent the normalized AUC. On one hand, the Table S5 will repeat the presentation of Figure 3, and on the other hand, in order to make the manuscript more concise, we thought to place the voluminous numerical data in the supplementary material. We hope that this answer suffices, however, we will comply with the reviewer’s decision if the reviewer still decides that Table S5 should be included in the main body of the manuscript.

Reviewer 2 Report
Thank you for your interesting research
INTRODUCTION are there more recent articles related?
Table 1 : Is there other possibility of presentation (too long)?
CONCLUSIONS According to data and references
Author Response
Dear reviewer, thank you for your comments and suggestions!
As far as the author’s knowledge, the most recent of relevant literature has been cited.
As for your suggestion, Table 1 has been reformatted so that it becomes more concise and less voluminous.

Reviewer 3 Report
The material presented in the manuscript is interesting. Using the analysis of morphological characteristics of several representatives of the genus Senecio and specialized plant metabolites makes it possible to assess their systematic position. Similar work was carried out at the end of the 20th century, but this study uses new methodological approaches. Comments are reflected in the manuscript.
Differences in the composition of phenolic compounds of various plant organs are known. The aerial parts of flowering plants were used in the work. It is possible that the established similarity/difference is a consequence of the quality of the plant material. This aspect should be taken into account when presenting the material, in the conclusion and in the abstract.

Author Response
Dear reviewer, thank you for your comments and suggestions! We hope the considerations taken will suffice your requirements.
Comment:
"Differences in the composition of phenolic compounds of various plant organs are known. The aerial parts of flowering plants were used in the work. It is possible that the established similarity/difference is a consequence of the quality of the plant material. This aspect should be taken into account when presenting the material, in the conclusion and in the abstract."
"What was the aerial part of the plants used for analysis (stem, leaf, flowers)?In what period of growth were the plants taken?
Which organs dominated in the sample (flowers or leaves)?"
Comment page 13
- What was the aerial part of the plants used for analysis (stem, leaf, flowers)?
Answer:
From each of the studied Senecioneae species, the aerial parts collected consisted of stem, leaves and flower heads (as detailed in Sections 3.1. and 3.2).
- In what period of growth were the plants taken?
Answer:
The plants were randomly collected at their natural habitats during full flowering stage in July 2021 (as detailed in 3.1. and 3.2.).
- Which organs dominated in the sample (flowers or leaves)?
Answer:
As wild growing plants, the aerial parts collected had their variability, with the raw data presented in Table S1, and summarized in Table S2. The coefficient of variation of all parameters typically was within 10-15 %CV, except X12, where the CV is around 30-35%. The phenolic content depends on the plant parts collected from the aerial parts.
Hence, the morphological variability indeed influenced the phenolic variability, however, the LC-MS measurements were performed on a combined sample from all 15 collected plants from each species, which provides the representative phenolic profile.
These considerations are commented in the revised manuscript as follows:
“As phenolic content naturally varies between different plant parts, the %CV of the morphometric characteristics recorded for each plant species, were typically below 20 %CV, except for number of capitula per plant (X12) reaching above 40 %CV (Table S2). Noteworthy, for each of the plant species, the LC-MS measurements on phenolic content were performed on homogenized samples from all 15 aerial plant samples, providing representative phenolic profile.”
